# Novel Insights on the Role of Epigenetics in Androgen Receptor’s Expression in Prostate Cancer

**DOI:** 10.3390/biom13101526

**Published:** 2023-10-14

**Authors:** Vânia Camilo, Mariana Brütt Pacheco, Filipa Moreira-Silva, Gonçalo Outeiro-Pinho, Vítor M. Gaspar, João F. Mano, C. Joana Marques, Rui Henrique, Carmen Jerónimo

**Affiliations:** 1Cancer Biology and Epigenetics Group, Research Center of IPO Porto (CI-IPOP)/RISE@CI-IPOP (Health Research Network), Portuguese Oncology Institute of Porto (IPO Porto)/Porto Comprehensive Cancer Center (Porto.CCC) Raquel Seruca, R. Dr. António Bernardino de Almeida, 4200-072 Porto, Portugal; van.cml@gmail.com (V.C.); mariana.brutt@gmail.com (M.B.P.); filipa.m.silva@ipoporto.min-saude.pt (F.M.-S.); goncalo.outeirodepinho@unibe.ch (G.O.-P.); henrique@ipoporto.min-saude.pt (R.H.); 2CICECO—Aveiro Institute of Materials, Department of Chemistry, University of Aveiro, Campus Universitário de Santiago, 3810-193 Aveiro, Portugal; vm.gaspar@ua.pt (V.M.G.);; 3Genetics Unit, Department of Pathology, Faculty of Medicine, University of Porto (FMUP), Alameda Prof. Hernâni Monteiro, 4200-319 Porto, Portugal; cmarques@med.up.pt; 4i3S-Institute for Research and Innovation in Health, University of Porto, R. Alfredo Allen 208, 4200-135 Porto, Portugal; 5Department of Pathology, Portuguese Oncology Institute of Porto (IPO Porto), Rua Dr. António Bernardino de Almeida, 4200-072 Porto, Portugal; 6Department of Pathology and Molecular Immunology, ICBAS-School of Medicine and Biomedical Sciences, University of Porto, Rua Jorge Viterbo Ferreira nº 228, 4050-313 Porto, Portugal

**Keywords:** prostate cancer, androgen receptor, DNA methylation, epigenetics, transcriptional repression

## Abstract

The androgens/androgen receptor (AR) axis is the main therapeutic target in prostate cancer (PCa). However, while initially responsive, a subset of tumors loses AR expression through mechanisms putatively associated with epigenetic modifications. In this study, we assessed the link between the presence of CpG methylation in the 5′UTR and promoter regions of AR and loss of AR expression. Hence, we characterized and compared the methylation signature at CpG resolution of these regulatory regions in vitro, both at basal levels and following treatment with 5-aza-2-deoxycytidine (DAC) alone, or in combination with Trichostatin A (TSA). Our results showed heterogeneity in the methylation signature of AR negative cell lines and pinpointed the proximal promoter region as the most consistently methylated site in DU-145. Furthermore, this region was extremely resistant to the demethylating effects of DAC and was only significantly demethylated upon concomitant treatment with TSA. Nevertheless, no AR re-expression was detected at the mRNA or protein level. Importantly, after treatment, there was a significant increase in repressive histone marks at AR region 1 in DU-145 cells. Altogether, our data indicate that AR region 1 genomic availability is crucial for AR expression and that the inhibition of histone methyltransferases might hold promise for AR re-expression.

## 1. Introduction

Prostate cancer (PCa) remains a major health concern, ranking second in incidence and fifth in mortality among all cancers in men, worldwide [1]. The androgens/androgen receptor (AR) axis is the main driver of PCa, since ligand-mediated AR activation triggers a signaling network that fuels PCa cell growth and survival [2]. This dependency on AR signaling is exploited therapeutically in androgen deprivation therapy (ADT), the standard-of-care for PCa patients with locally advanced or metastatic disease. Nevertheless, although initially successful, resistance to ADT frequently occurs in the form of castration resistant prostate cancer (CRPC) [3]. Importantly, as CRPC becomes independent of circulating androgens, the great majority remain dependent on AR signaling. The mechanisms contributing to this effect include AR overexpression, mutations in the ligand-binding domain (LBD), the emergence of constitutively active splice variants, intra-tumoral androgens’ biosynthesis or upregulation of AR co-activators [4]. However, in some CRPCs, there is an extensive loss of AR expression, and in rare cases, complete loss is reported [5]. These tumors pose a therapeutic challenge since they are inherently more resistant to conventional therapies. Furthermore, AR-independent CRPCs are increasingly prevalent, which may result from the selective pressure arising from the use of more potent AR-targeted therapies [6]. Hence, clarification of the mechanisms underlying AR loss are needed to better address this clinical challenge. One possible contributor is aberrant DNA methylation, which has been shown to contribute to cancer development and progression [7,8,9].

The human AR gene is located at Xq11.2-q12, encoding a 10.6 kb transcript that comprises a 1.1 kb long 5′untranslated region (5′UTR) that contains a 24 bp upstream open reading frame (uORF), a 2.7 kb coding region and a 6.8 kb 3′ untranslated region (3′UTR) [10]. Deletion mapping and site-directed mutagenesis assays locate the AR core promoter to a region between −74 and +87 bp in relation to the transcription starting site (TSS) [11]. This promoter region lacks TATA and CCAAT boxes, with transcription relying on recruitment of zinc finger transcription factor specificity protein 1 (Sp1) to a GC box located −46 to −41 bp (Sp1-1) from the TSS. When bound, Sp1 associates with transcriptional machinery (TFIID) to facilitate RNA polymerase II-mediated transcription. Importantly, Sp1 also acts as a recruiter of chromatin remodelers and histone modifying enzymes [12]. Additional GC regulatory boxes with putative regulatory importance have been described throughout the 5′UTR [13,14], as well as additional sites for specific binding of transcription factors, which may promote or repress AR transcription [15].

The hypothesis of promoter CpG hypermethylation as a mechanism for AR gene inactivation was previously documented [16,17,18]. In these studies, a 3 kb CpG island was identified, spanning from the promoter region to the first exon [18]. Importantly, in some AR negative PCa cell lines, aberrant methylation of this CpG island was associated with transcriptional silencing. This methylation signature was never reported for AR-expressing PCa cell lines or normal prostate tissues [16,17]. Detailed CpG methylation analysis of AR positive and negative cell lines led to the identification of two main methylation hotspots, the first in a region adjacent to the pur/pyr rich stretch (−131, −125, −123 and −121 bp upstream from TSS) and the second within the core promoter, in 5′UTR region (+44, +49 and +54 bp downstream from the TSS) [16,19].

The observation of a methylation-related silencing mechanism prompted several attempts to reverse methylation by exposing AR negative cell lines to demethylating drugs, for relatively short or prolonged periods of time, alone or in combination with histone deacetylating drugs. Jarrard et al. showed that treatment with low doses of the demethylating drug 5-aza-2-deoxycytidine (DAC) for a short period of time was able to restore AR mRNA expression in some AR negative cell lines but not in DU-145 cells, in which AR expression was not recovered even after chronic exposure for 2 months [18]. A similar absence of effect was reported in two other studies after short-term exposure to DAC [20,21]. On the other hand, Nakayama et al. demonstrated that exposure of DU-145 to DAC alone or in combination with Trichostatin A (TSA) led to successful AR transcript expression. Intriguingly, TSA treatment was more effective than DAC in promoting AR expression, which led the authors to propose that methylation might only partially impact AR expression [17].

Nonetheless, methylation alone does not seem to fully explain AR loss, since some AR negative cells lines and clinical specimens with heterogeneous AR staining do not exhibit this dense methylation signature [22]. Therefore, alternative mechanisms involving several transcriptional regulatory regions [15], post-transcriptional regulation via miRNAs and RNA binding proteins, as well as post-translational regulation of AR via proteasome-mediated degradation and histone modifying enzymes [23], may also contribute to AR silencing [24].

Herein, we sought to further dissect the role of promoter methylation for AR transcriptional silencing. Thus, we investigated the methylation signature of prostate cell lines with different AR transcript expression profiles at CpG resolution. CpG methylation assessment in basal conditions and following treatment with DAC alone or in combination with TSA allowed us to better understand which sites are most likely to influence AR transcriptional regulation. Importantly, a deeper analysis of chromatin architecture upon DAC/TSA treatment revealed a highly repressed AR region 1, which partially explains why DNA methyltransferases targeting alone has led to ineffective results in AR re-expression and raising the possibility of targeting histone methyltransferases as an alternative approach.

## 2. Materials and Methods

### 2.1. Cell Culture

In this study, AR positive (LNCaP and 22Rv1) and AR negative (DU-145 and PC-3) human PCa cell lines, as well as the non-malignant prostate cell line RWPE-1 were used [25]. LNCaP, 22Rv1 and PC-3 cells were grown in RPMI 1640 media (PAN-Biotech, Aidenbach, Germany); DU-145 was grown in MEM (PAN-Biotech, Aidenbach, Germany); and RWPE-1 cells were grown in K-SFM basal media (GIBCO, Invitrogen, Waltham, MA, USA). All media were supplemented with 10% (*w*/*v*) heat-inactivated fetal bovine serum (FBS; Biochrom, Merck, Darmstadt, Germany) and 1% (*w*/*v*) penicillin/streptomycin (Grisp, Portugal). Cells were maintained in an incubator at 37 °C and 5% CO_2_. All cell lines used in this study were routinely tested for *Mycoplasma* spp. contamination by using a TaKaRa PCR Mycoplasma Detection Set (Clontech Laboratories, Mountain View, CA, USA).

### 2.2. DAC and TSA Treatments

For DAC (Sigma-Aldrich, St Louis, MO, USA), a 100 mM stock solution was prepared in acetic acid and stored at −80 °C until further use. DAC working solutions of 1 mM were freshly prepared by further dilutions in DPBS along with a vehicle control (1% acetic acid in DPBS). For TSA (Sigma-Aldrich, St Louis, MO, USA), a 5 mM stock solution was diluted 10× to a working concentration of 0.5 mM in DMSO.

For each experiment, 5 × 10^5^ cells were platted in T-25 cm^2^ culture flasks and treated daily for 3 days with 5 µM of DAC [26] or its vehicle, alone and/or in combination with 0.33 µM TSA [27] or DMSO. The culture media was renewed daily with each treatment.

### 2.3. Western Blot Analysis

Total protein extracts were prepared in RIPA Lysis Buffer (Santa Cruz Biotechnology, Dallas, TX, USA) containing a protease inhibitor cocktail (Roche, Basel, Switzerland), sonicated and centrifuged at 13,000 rpm for 30 min at 4 °C. Protein concentration was determined using the Pierce BCA Protein Assay Kit (Thermo Scientific Inc., Waltham, MA, USA), according to the manufacturer’s instructions. For AR detection, 20–50 μg of total protein was used for SDS-PAGE, as previously described [28]. The protein was loaded into an 8% polyacrylamide and transferred to a nitrocellulose membrane using a Trans-Blot Turbo Transfer system (Bio-Rad, Hercules, CA, USA). Membranes were blocked with 5% bovine serum albumin (BSA; Santa Cruz Biotechnology, Dallas, TX, USA) in TBS-0.1% (*w*/*v*) Tween for 1 to 2 h at room temperature. Afterwards, the membranes were incubated overnight at 4 °C with the respective primary antibodies (Appendix A). On the following day, the membranes were washed and incubated with the appropriate HRP-labeled secondary antibody (1:5000 dilution in 5% (*w*/*v*) milk/TBS-0.1% (*w*/*v*) Tween) (Bio-Rad, Hercules, CA, USA) for 1 h at room temperature. The signal was developed with an enhanced chemiluminescence detection kit (Clarity and Clarity Max ECL Western Blotting Substrates (Bio-Rad, Hercules, CA, USA) [29]. β-actin was used as the loading control.

### 2.4. DNA Extraction and Bisulfite Modification

Genomic DNA was obtained by digestion with proteinase K (20 mg/mL) at 55 °C in the presence of 10% (*w*/*v*) SDS, followed by extraction using the phenol/chloroform method and precipitation with 100% (*w*/*v*) ethanol, as described by us [28]. Bisulfite modification of 1 µg of DNA was performed using the EZ DNA Methylation Gold Kit (Zymo Research, Irvine, CA, USA), according to the manufacturer’s suggestions [29].

### 2.5. Bisulfite Sequencing

AR promoter region was divided into three subregions (region 1, region 2 and region 3) that were amplified using primer pairs designed using Methyl Primer Express v1.0 (Appendix A) and Xpert Hotstart Mastermix (Grisp, Porto, Portugal). The specificity of the PCR reaction was assessed in 2% (*w*/*v*) agarose gels before proceeding to the cloning step. Two μL of the PCR product were subcloned into the TOPO TA vector (TOPO TA Cloning kit for sequencing, Invitrogen, Waltham, MA, USA) and used for transformation of One Shot^®^ chemically competent *E. coli* vials (subcloning efficiency DH5α competent cells, Invitrogen, Waltham, MA, USA) by heat-shock at 42 °C for 45 s, as adapted from [30]. Following recuperation in S.O.C., (NzyTech, Lisbon, Portugal) for 1 h, the bacteria were seeded in pre-warmed LB plates containing 50 µg/mL ampicillin (NZytech, Lisbon, Portugal) and 40 ng/mL X-Gal (NzyTech, Lisbon, Portugal) and incubated overnight at 37 °C. The next day, at least 10 white colonies were collected and subjected to colony PCR using Xpert Hotstart Mastermix (Grisp, Porto, Portugal) and M13 forward and reverse primers to confirm DNA insertion. The DNA was purified using Illustra GFX PCR DNA and Gel Band Purification Kit (GE Healthcare, Chicago, IL, USA) according to manufacturers’ instructions and used to perform the sequencing reaction using the BigDye Terminator v3.1 Cycle Sequencing kit (Applied Biosystems™, Waltham, MA, USA) using either a M13 forward or reverse primer. The products were purified using Sephadex 50 resin (GE Healthcare, Chicago, IL, USA) and 20 µL of formamide (Sigma-Aldrich, St Louis, MO, USA) was added, followed by a denaturing step at 95 °C for 5 min. Finally, the samples were sequenced by Sanger sequencing in a 3500 Genetic Analyzer (Applied Biosystems™, Waltham, MA, USA).

QUMA (http://quma.cdb.riken.jp/, last accessed on 12 November 2022) software was used to analyze bisulfite sequencing data [31]. For each cell line and region, the methylation status of 15 alleles was assessed by sequencing 15 clones per condition, obtained from bisulfite treated DNA. The percentages of minimum sequence identity and minimum conversion rate were 90% and 95%, respectively.

### 2.6. Quantitative Methylation Specific PCR (qMSP)

Bisulfite modified DNA was amplified by qMSP using Xpert Fast SYBER Mastermix Blue (GRiSP, Porto, Portugal) and specific primers (Appendix A), as previously reported by us [29]. *β-actin* was used as reference gene. For each sample, the methylation levels were obtained from a calibration curve that was made by using serial dilutions of a bisulfite modified Human HCT116 DKO methylated DNA standard (Zymo Research, Irvine, CA, USA) following normalization to *β-actin*. For each gene, both an NTC and a bisulfite modified HCT116 DKO non-methylated DNA standard (Zymo Research, Irvine, CA, USA) were used as negative controls.

### 2.7. Chromatin Immunoprecipitation

T-175 cm^2^ culture flasks containing 90% confluent DU-145 cells from each experimental condition were used for ChiP-PCR experiments, based on a protocol previously established by us [32]. For that, DU-145 cells were crosslinked by adding formaldehyde directly to the flasks (to a 1% (*w*/*v*) formaldehyde final concentration). The flasks were then incubated at room temperature (RT) for 10 min with constant agitation, after which the reaction was quenched by the addition of glycine (final concentration of 0.125 M), followed by incubation at RT for 5 additional minutes with agitation. Crosslinked cells were then washed twice with ice cold DPBS before 10 mL of ice cold DPBS containing a protease inhibitor cocktail (Roche, Basel, Switzerland) was added to each flask and a rubber cell scraper was used to collect cells. A centrifugation step (4 °C) at 800× *g* allowed for cell collection. A cell lysis buffer (10 mM Tris-HCl pH7.5, 10 mM NaCl 0.5%, NP-40) containing a protease inhibitor cocktail (Roche, Basel, Switzerland) was added to the cell pellet and lysis was performed for 1 h 30 min on ice, with intermittent vortexing. A centrifugation step (4 °C) at 800× *g* that allowed nuclei collection was followed by incubation on ice for 15 min with an ice-cold nuclei lysis buffer (50 mM Tris-HCl pH7.5, 10 mM EDTA pH8, 1% (*w*/*v*) SDS) containing a protease inhibitor cocktail (Roche, Basel, Switzerland). The supernatant was diluted with 2× volume of IP dilution buffer (16.7 mM Tris-HCl pH7.5, 167 mM NaCl, 1.2 mM EDTA pH8, 0.01% (*w*/*v*) SDS) before sonication for 2 × 10 min cycles of 30 s ON and 30 s OFF, replacing warm with cold water and ice between each cycle. Sonicated samples were centrifuged at 13,000 rpm for 10 min at 4 °C and the supernatant (containing chromatin) was collected in a new 1.5 mL tube. Thirty µg of chromatin was set aside to assess shearing efficiency. An input fraction was set aside, and the sheared chromatin was diluted 10x in ChIP dilution buffer (16.7 mM Tris-HCl pH8.1, 167 mM NaCl, 1.2 mM EDTA pH8, 0.01% (*w*/*v*) SDS, 1.1% (*w*/*v*) TritonX-100). The final volume was divided in 3 parts, to which 20 µL of (Dyna)Beads A + G (16-663, EMD, Millipore, Burlington, MA, USA) plus either (mono)clonal anti-acetyl histone 3 antibody or (mono)clonal anti-RNA polymerase II (positive control) or mouse/rabbit IgGs (negative control) were added. The immunoprecipitation was performed overnight at 4 °C. The next day, the (Dyna)Beads were successively washed for 5 min at 4 °C, with agitation with 1 mL of (i) low salt wash buffer (20 mM Tris-HCl pH8, 2 mM EDTA, 150 mM NaCl, 0.1% (*w*/*v*) SDS, 0.1% (*w*/*v*) Triton X-100); (ii) high salt wash buffer (20 mM Tris-HCl pH8, 2 mM EDTA, 500 mM NaCl, 0.1% (*w*/*v*) SDS, 0.1% (*w*/*v*) Triton X-100); (iii) LiCl wash buffer (0.5 M LiCl, 100 mM Tris-HCl pH9, 1% (*w*/*v*) NP-40, 1% (*w*/*v*) deoxycholate); and (iv) TE buffer (10 mM Tris-HCl pH8, 1 mM EDTA). The DNA was eluted by adding 100 µL of elution buffer (1% (*w*/*v*) SDS, 0.1 M NaHCO_3_) to the beads in the presence of 0.5 µL of RNase A and incubated for 30 min at 37 °C with agitation. The input fraction was subjected to the same conditions. Afterwards, treatment with proteinase K was performed for 2 h at 62 °C, with agitation. The crosslinking was reversed by placing the samples at 95 °C for 10 min.

The supernatant was then collected, and the DNA was purified using the Qiaquick extraction kit (Qiagen, Hilden, Germany) according to the manufacturer’s instructions, followed by RT-qPCR with four pairs of primers (Appendix A and Appendix A). Data shown are presented using the formula %Input [100 × (2^(CT Raw mean − CT))]. Normal mouse IgG and RNA polymerase II protein immunoprecipitation were used as internal controls.

### 2.8. RNA Extraction, cDNA Synthesis and RT-qPCR

RNA extraction was performed using TripleXtractor (GRiSP, Porto, Portugal) according to the manufacturer’s instructions. Afterwards, 1 µg of RNA was converted to cDNA using a RevertAid RT kit (Thermo Scientific Inc., Waltham, MA, USA) [28]. Quantitative real time PCR reactions were performed in an Applied Biosystems 7500 Real-Time PCR System using TaqMan^®^ gene expression assays (Thermo Scientific Inc., Waltham, MA, USA) for *AR* (assay ID Hs00171172_m1) and housekeeping gene *β-GUS* (assay ID Hs99999908_m1). LNCaP and 22Rv1 cDNAs were used as positive controls. Three technical replicates were made for each biological replicate and results were normalized to *β-GUS*. Fold-changes to mRNA expression were calculated using the cycle threshold (ΔΔCt) method [28].

### 2.9. Statistical Analysis

Statistical analysis was performed using the GraphPad Prim 7.0 software (GraphPad Software Inc., San Diego, CA, USA). Non-parametric Mann–Whitney *U* test was used to compare two groups. For comparisons between three or more groups, non-parametric Kruskal–Wallis test, with Dunn’s correction, was used. All in vitro experiments were performed at least 3 times.

Moreover, QUMA software Fisher’s exact test was used to determine the statistical significance of methylation levels between two bisulfite sequence groups at each CpG, whereas the Mann–Whitney *U*-test was used for the evaluation of an entire set of CpG sites.

*p*-values were considered statistically significant when inferior to 0.05. Significance is shown vs. the respective control and depicted as follows: * *p* < 0.05, ** *p* < 0.01, *** *p* <0.001; **** *p* < 0.0001. All data are represented as mean ± standard deviation of the mean (SEM), unless otherwise specified.

## 3. Results

### 3.1. Methylation Profiling of AR Negative Prostate Cancer Cell Lines

The methylation status of individual CpGs, present in regulatory regions of the *AR* gene of both AR negative (DU-145 and PC-3) and positive (RWPE-1) cell lines, was analyzed by bisulfite sequencing. Based on prior reports [16,17,19], three regions were defined, as indicated in Figure 1A. Region 1, located ~250 bp upstream from the TSS, covers most of the core promoter region (−74 to +87) [11] and it encompasses 16 CpGs sites (from CpG at −259 to +20), including the one at the Sp1-1 binding site (5′-GGGGCGGG-3′; −42) [33]. Region 2 is contiguous with region 1 and located in the 5′UTR. It covers 21 CpGs (from CpG at +44 to +264). Finally, region 3 is located on the first exon and contains 14 CpGs (from CpG at +1162 to +1408).

We observed methylation heterogeneity of CpG sites across regions and clones (Figure 1B and Appendix A). In DU-145, CpGs surrounding the core promoter and the TSS (regions 1 and 2) were densely methylated in all alleles (Figure 1B). In region 1, the percentage of methylation per CpG site was always above 50%, with CpGs −131, −30 and +20 reaching 93.3% methylation. However, these CpGs were also more frequently methylated in AR positive cell line RWPE-1, in which the mean percentage methylation in region 1 was 20% (Figure 1C) and all the CpGs had individual methylation levels above this percentage (53.3%, 40% and 33.3%, respectively, Figure 1B). Furthermore, similarly to the other CpGs, the putative methylation hotspot (−131 to −121 bp from TSS) was mostly unmethylated in RWPE-1 cells (Figure 1B).

Conversely, in DU-145 clones, the least frequently methylated CpG site in region 1 was that embedded in the Sp1-1 binding site (−42) (unmethylated in 7/15 clones, 53.3% methylation). Furthermore, the CpGs located near −42 were also more heavily methylated in DU-145 than in the other cell lines. Of note, CpG −51, which was seldom methylated in PC-3 (6.7%) and RWPE-1 (0%), was the CpG with the highest difference in percentage of methylation between DU-145 and the other cell lines (Figure 1B).

For region 2, DU-145 exhibited, again, the highest overall methylation of all cell lines (66.3%, Figure 1C). However, the methylation levels were inferior to those observed for region 1 (80.0%) and not so distant from those disclosed for PC-3 (38.5%) and RWPE-1 (45.4%) (Figure 1C). For all cell lines, CpG +145 was always unmethylated and, for DU-145 and RWPE-1, this was also the case of CpG +159, which was only found methylated in 1/15 clones in PC-3 cells (6.7%). Additionally, CpGs +80 and +264 were also unmethylated in RWPE-1 but depicted some degree of methylation in DU-145 and PC-3 cells. Interestingly, although CpG + 80 exhibited lower overall methylation levels than other CpGs in region 2, this CpG site disclosed the highest difference in methylation for both DU-145 (53.3%) and PC-3 (20.0%) comparatively to RWPE-1 (Figure 1B).

Importantly, whereas DU-145 cells presented high levels of methylation in both region 1 and 2 for almost all of the clones (13/15 clones with methylation levels ~75%), for both regions the clones obtained from PC-3 cells could be separated into two opposite groups: those with slight (~30%) or no methylation across the entire region and those with high methylation levels (above 75%). The same was observed for region 2 in RWPE-1 cells, in which 6/15 clones had methylation levels above 75% (Appendix A). This suggests that variability in methylation patterns was already present at the basal level in all cell lines but was considerably less frequent in DU-145.

Globally, for all the cell lines, regions 1 and 2 were more densely methylated than region 3 (Figure 1B,C). The DU-145 cell line was the most densely methylated across regions. PC-3 cells had similar percentages of methylation for both regions 1 and 2, whereas RWPE-1 was more methylated in region 2. For all cell lines, methylation in region 3 was either absent or extremely low (Figure 1B). Importantly, low methylation levels throughout regions 1 and 2 were observed for AR positive PCa cell lines LNCaP and 22Rv1 (Appendix A). Thus, we focused on regions 1 and 2 for further studies.

### 3.2. Methylation Profiling following Treatment with Demethylating Agent 5-Aza-2-Deoxyxitidine (DAC)

We next focused on characterizing the methylation status of individual CpG sites in regions 1 and 2 following exposure to the demethylating drug DAC. Bisulfite sequencing was performed in DNA obtained from treated cell lines (Figure 2A and Appendix A). Overall, DU-145 cells treated with vehicle conditions exhibited methylation levels similar to those obtained for WTs in both regions (Figure 2A). This was also observed for region 1 in RWPE-1 cells (Figure 2A). However, PC-3 vehicle-treated cells displayed a lower level of methylation across regions 1 and 2 in comparison with WT cells, and this was observed for RWPE-1 cells in region 2 (Figure 2A). Based on our previous observations, this seemed to be related to the inherent methylation variability among clones and was not likely to be derived from exposure to the vehicle solution.

In DU-145, treatment with DAC only accomplished a reduction in overall methylation of region 2, whereas region 1 was particularly resistant to DAC-mediated demethylation (Figure 2A,B). Similarly, for PC-3 and RWPE-1 cell lines, DAC treatment did not further decrease the methylation across regions. In fact, for region 1, DAC treatment associated with a slight increase, albeit non-significant, in the methylation levels of region 1 in PC-3 and in both regions 1 and 2 in RWPE-1 cells (Figure 2B). A closer inspection of the two proposed methylation hotspots revealed a 20–25% drop in the methylation of CpGs located at the 5′UTR (+44, +49 and +54 in region 2). However, the methylation of CpGs that comprised the promoter methylation hotspot (−131, −125, −123 and −121 in region 1) were largely unaffected by treatment (Figure 2A).

To further demonstrate that the lack of an obvious demethylation effect following DAC treatment was not due to the absence of drug action, methylation levels of *miR-130a* [34], *CCND2* [35] and *RASSF1A* [36], previously shown to be regulated by promoter methylation, were assessed by qMSP. Overall, a significant demethylation was observed for all genes tested upon DAC exposure (Figure 2C), validating the DAC treatment protocol.

Finally, we checked whether AR transcript and protein expression could be rescued by DAC treatment alone. However, no AR expression, either at mRNA or protein levels (Figure 2D), was observed.

### 3.3. Methylation Profiling following Concomitant Treatment with Demethylating Agent 5-Aza-CdR (DAC) and Histone Deacetylase Inhibitor Trichostatin A (TSA)

Since synergy is frequently observed in epigenetic mechanisms, concomitant treatment with DNA methyltransferase inhibitor (DNMTi) and histone deacetylase inhibitor (HDACi) was performed to investigate whether the HDACi TSA would enhance the DAC demethylating effect. Since DU-145 was more packed with methylation throughout regions 1 and 2, we focused on this cell line for the assay. To our knowledge, this is the first study analyzing the CpG site specific methylation following this combined approach.

Remarkably, TSA clearly potentiated DAC’s effect on region 1, while no effect was apparent on region 2 (Figure 3A,B). Interestingly, despite the overall demethylation on region 1, methylation was present in at least half of the alleles in 8/16 CpG sites, and 2/4 CpGs in the hotspot (−123, −121) were the more frequently methylated sites (64.3%, Figure 3A). However, no significant alteration of *AR* expression was found upon exposure to this combined treatment, both at transcript and protein levels (Figure 3C).

### 3.4. 5-Aza-CdR (DAC) and Trichostatin A (TSA) Concomitant Treatment Increases Repressive Histone Marks around AR Region 1

Following the previous data, we hypothesized whether the unsuccessful attempts to re-express AR upon DAC and TSA treatment were due to chromatin compactness across the *AR* promoter region. Thus, DU-145 cells were concomitantly treated with DAC and TSA, and the chromatin architecture of region 1 was analyzed by ChIP-PCR. Based on the literature [37], four repressive marks (H3K9me3, H3K27me3, H3K9me2 and H4K20me3) and three activating marks (H3K9Ac, H3K27Ac and H3K36me2) were selected for testing.

Indeed, upon treatment, a significant increase in repressive marks on *AR* region 1 (Figure 4A), as well as a decreasing trend for activating ones (Figure 4B), was observed, suggesting that a compact chromatin architecture might be impairing AR re-expression after treatment (Figure 4C).

## 4. Discussion

Treatment of locally advanced/metastatic PCa patients is mainly achieved by blocking androgen’s biosynthesis and/or hindering its binding to AR. Despite initial clinical benefit, patients become refractory to this therapy, mostly due to alterations in *AR* structure or expression [22,37]. While most of these tumors remain dependent on AR signaling, resistance to therapy in the form of AR-low or negative tumors, with or without evidence of neuroendocrine differentiation, is increasing in prevalence due to the growing inefficacy and selectivity of AR-targeted therapies [5]. These tumors are therapeutically challenging due to the lack of durable response to chemotherapy and/or of alternative treatment options [38]. Hence, illuminating the mechanisms that contribute to AR loss is of paramount clinical significance.

Promoter CpG island hypermethylation has been classically associated with transcriptional silencing [39]. Initial findings of a negative correlation between the presence of promoter CpG methylation and AR expression, both in cell lines and clinical samples, have supported this mechanism as a contributing factor for AR transcriptional silencing [16,18]. Subsequent studies have attempted to pinpoint the specific regions and CpG sites within the *AR* promoter and 5′UTR that might be responsible for this effect. In two different studies, two hotspots in the proximal promoter (−131 to −121 bp from TSS) and 5′UTR (+44 to +54 bp from the TSS) were reported as critical for *AR* expression, mostly based on comparison to methylation patterns of AR positive cells and on recovery of AR transcript expression following exposure to DAC, alone or in combination with TSA or sodium butyrate (NaB) [17,19,20,40]. In line with previous reports, we found extensive methylation throughout *AR* regulatory regions in DU-145. Surprisingly, DAC treatment did not significantly impact CpG methylation at the proximal promoter region, and only minor effects were observed for 5′UTR hotspots. Importantly, significant demethylation of the promoter region of *AR* was only observed upon DAC and TSA concomitant treatment, but the CpG hotspots retained high methylation levels, especially in CpGs at −123 and −121 bp from the TSS.

Our results further confirm that not all AR negative cells exhibit dense methylation in these same regions, indicating that methylation-independent mechanisms also influence *AR* expression. Indeed, blocked translation, and not transcription, has been suggested for AR-negative cell line PC-3, which bears much lower methylation levels across those regions than DU-145 [19]. In line with this, a 180 bp region, spanning from +21 to +149 bp, that includes a stem-loop secondary structure at +109 to +129 bp, fundamental for successful translation, has been proposed as the problematic site [19]. However, we found that PC-3 cells exhibited *AR* gene methylation heterogeneity, especially in region 1, with some of the sequenced alleles displaying extensive methylation in almost all CpGs. This may indicate that, within the same tumor, cells may use different mechanisms to inactivate AR expression, some being DNA methylation-dependent while others not. Indeed, other epigenetic mechanisms, particularly histone methylation, might further explain the lack of AR expression in PC-3 cells [41]. Advances in the field of single-cell epigenomics [42] may help shed light on this question, and novel techniques, such as coupling microscopy-based epigenetic visualization assay (EVA) to RNA-FISH, may allow the simultaneous analysis of epigenetic and transcriptional profiles [43]. Additionally, recent epigenome-editing techniques may help to elucidate which CpG sites might be more directly controlling AR expression [44,45,46].

Traditionally, methylation is perceived as a chemical repellent for transcription factor (TF) binding. However, recent studies challenge this observation by showing that TFs respond differently to the presence of DNA methylation [47]. In this vein, Sp1, which is essential for *AR* transcription, is a methylation indifferent transcription factor, and methylation in CG boxes is not expected to prevent DNA binding [48]. Indeed, AR negative cells maintain the Sp1-1 binding site relatively unmethylated in comparison with the neighbor CpGs. Nonetheless, it is important to highlight that Sp1’s interactome is vast and that its described partners include epigenetic modulators with opposite effects on gene transcription [49]. For instance, Sp1 is known to recruit PRMT5, which then associates with ATPase-dependent chromatin remodeler Brg1 in the proximal promoter region of AR [50], also functioning as a platform for the recruitment of transcription repressors such as DNMT1, G9a and HDAC1 [51].

Importantly, our ChIP-PCR data corroborate this argument, since a heavily packed chromatin architecture around *AR* region 1 was found, possibly impairing the binding of TF and transcription itself, which might explain the differences observed between our study and the results of Nakayama et al. [17]. Moreover, prostate tumor cells frequently present aberrant patterns of both DNA methylation and histone post-translational modifications [19], which is in line with our ChiP-PCR results. Furthermore, several chromatin remodeling enzymes are overexpressed in CRPC, and might regulate AR expression, in association with its promoter methylation [52]. With an ever-growing number of articles showing the relevance of histone-modifying enzymes on CRPC cells [28], our data further sustain the use of inhibitors against such enzymes [53], alone or in combination with others, as an alternative for attempting AR re-expression.

Overall, our findings pave the way for the use of histone methyltransferase inhibitors (HMTi) and/or DNMTi, in combination with ADT, for the management of CRPC. Indeed, in AR-negative tumors, epigenetics might explain the lack of the receptor’s expression. Even thought we did not manage to re-express AR in DU-145 CRPC cells, in primary tumors, due to intra-tumor and inter-patient heterogeneity, the possibility of reverting AR promoter repressive chromatin state may induce AR expression. Therefore, combining epigenetic drugs with the standard of care might improve PCa patients’ management [53].

However, before clinical implementation, the rationale of this study would need to be performed with additional AR-negative cell lines to enlarge the spectrum of in vitro models to account for tumor heterogeneity and strengthen the results. Moreover, our current work lacks validation in patient-derived materials, as well as validation of AR methylation by qMSP, since we were not able to optimize specific primers. Additionally, a ChiP-seq approach might give a more in-depth knowledge of the chromatin architecture nearby the AR promoter, and the usage of AR specific antibodies would allow to further support our hypothesis. Importantly, studies involving HMTi and DNMTi are still required to fully address this question and validate the relevance of the epigenetic mechanisms for AR expression, which could ultimately provide clinical benefit for patient management.

## Figures and Tables

**Figure 1 biomolecules-13-01526-f001:**
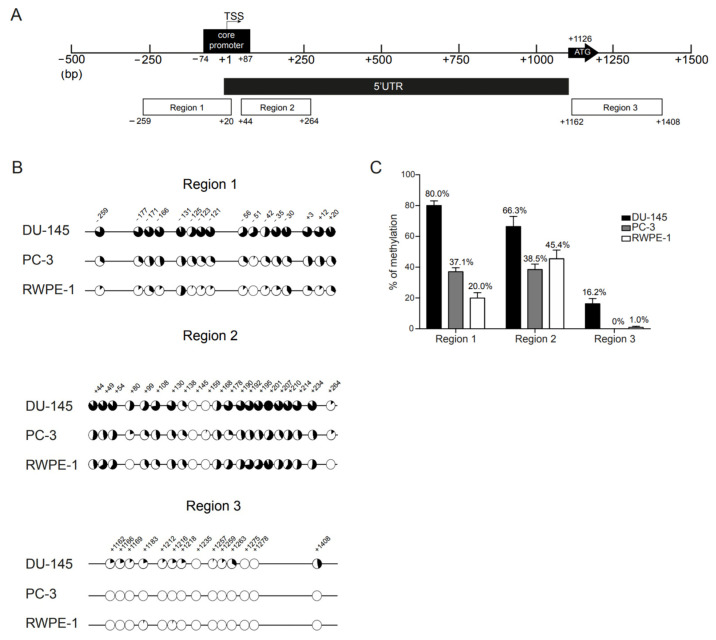
Methylation profile of *AR* regulatory regions in prostate cancer cell lines. (**A**) Schematic representation of the human *AR* gene 5′UTR and proximal promoter regions. The regions examined by bisulfite sequencing cover ~250 bp upstream the TSS (region 1) and the core promoter region (−74 to +87 bp) (regions 1 and 2), the first 250 bp of the 5′UTR (region 2) and the region surrounding the translation start site (region 3). The first and last CpG sites of each region are found under each respective region box. The bent arrow marks the transcription start site (TSS; +1) and the ATG arrow represents the translation start at +1126 bp (exon 1). The CpG sites were numbered according to previous reports [14]. (**B**) Mean percentage of methylation per CpG across regions 1 to 3, in DU-145 (upper), PC-3 (middle) and RWPE-1 (lower) cell lines. Closed circles represent 100% methylation, whereas open circles represent 0% methylation. (**C**) Overall percentage of methylation per region obtained for each cell line. The percentage of methylation is obtained by dividing the total number of methylated CpG sites within a region by the total number of CpG sites that are present in that region. Data are represented as mean ± standard error of the mean (SEM).

**Figure 2 biomolecules-13-01526-f002:**
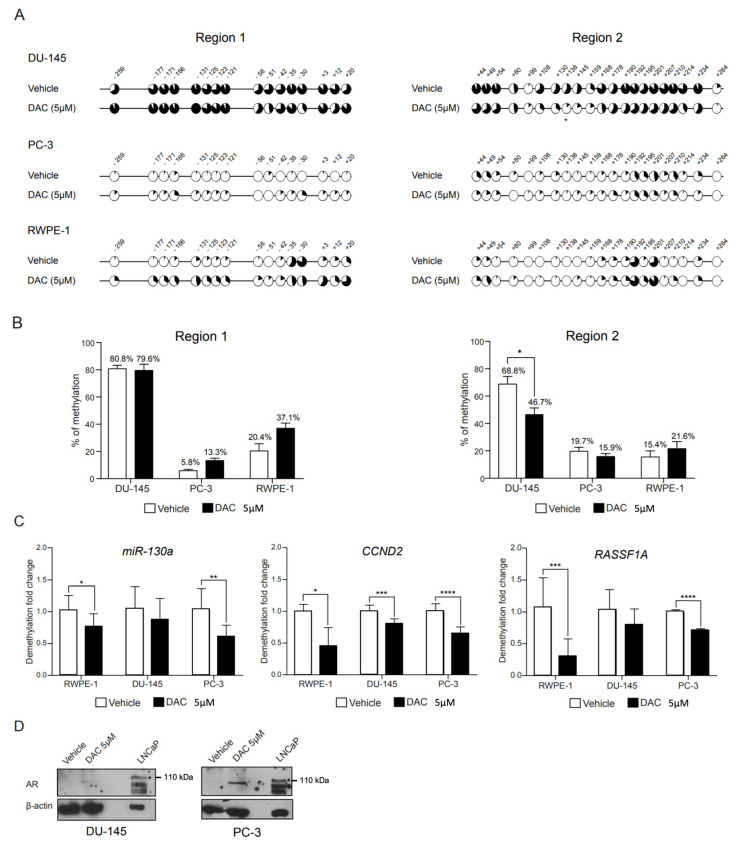
Methylation profile of AR regulatory regions 1 and 2 following DAC treatment. (**A**) Mean percentage of methylation per CpG across regions 1 and 2, in DU-145 (upper), PC-3 (middle) and RWPE-1 (lower) cell lines following treatment with DAC at 5 µM or its respective vehicle. (**B**) Overall percentage of methylation per region obtained for each cell line. Data are represented as mean ± standard error of the mean (SEM). (**C**) Methylation levels of *miR-130a*, *CCND2* and *RASSF1A* genes in prostate cell lines treated with DAC at 5 µM or its respective control. (**D**) Western blot of AR following 72 h exposure to DAC at 5 µM or its respective control. An asterisk (*) marks the specific AR full length band at 110 kDa. (Significance level represented as: * *p* < 0.05; ** *p* < 0.01; *** *p* < 0.001; **** *p* < 0.0001.)

**Figure 3 biomolecules-13-01526-f003:**
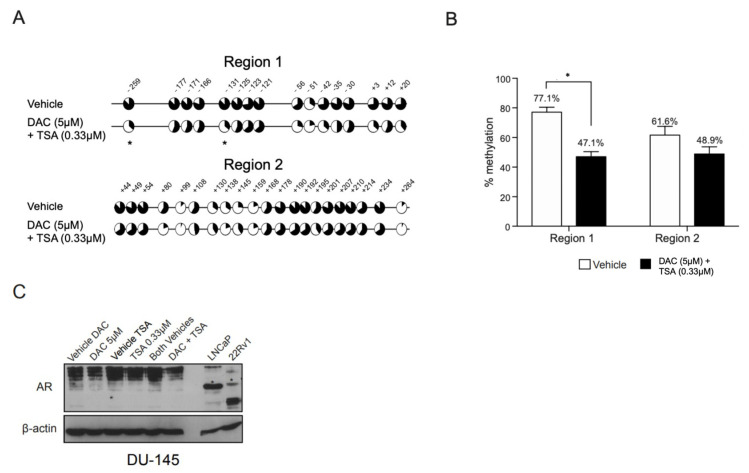
Methylation profile of AR regulatory regions 1 and 2 in DU-145 cells following combined treatment with DAC and TSA. (**A**) Mean percentage of methylation per CpG across regions 1 and 2 in DU-145 following combined treatment with DAC at 5 µM and TSA at 0.33 µM or their respective vehicles. (**B**) Overall percentage of methylation per region. Data are represented as mean ± standard error of the mean (SEM). (**C**) Western blot of AR following 72 h exposure to DAC at 5 µM and TSA at 0.33 µM, both alone and in combination, and the respective vehicles. An asterisk (*) marks the specific AR full length band at 110 kDa. (Significance level represented as: * *p* < 0.05.)

**Figure 4 biomolecules-13-01526-f004:**
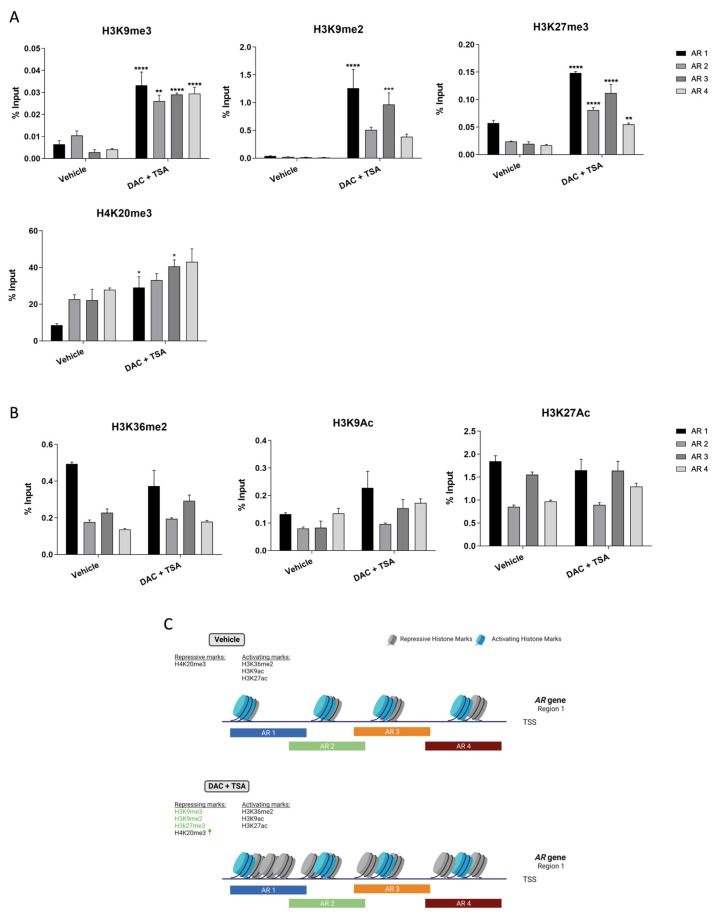
Chromatin architecture of DU-145 cells at *AR* region 1 following DAC/TSA treatment. (**A**) Mean percentage of histone repressive marks H3K9me3, H3K9me2, H3K27me3 and H4K20me3, as well as activating marks. (**B**) H3K36me2, H3K9Ac and H3K27Ac following concomitant treatment with DAC at 5 µM and TSA at 0.33 µM. (**C**) Schematic representation of the chromatin architecture of DU-145 cells after DAC/TSA treatment. ↑ represents increased levels of the respective hisytone mark. Created with Biorender.com (Accessed on 8 February 2023). (Significance level represented as: * *p* < 0.05, ** *p* < 0.01, *** *p* <0.001; **** *p* < 0.0001.)

## Data Availability

The data presented in this study are available in this article (and Appendix A).

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
