# Peer review of "Novel Insights on the Role of Epigenetics in Androgen Receptor’s Expression in Prostate Cancer"

_biomolecules, 2023, doi:10.3390/biom13101526_

Round 1

Reviewer 1 Report

Biomolecules

Novel insights on the role of epigenetics in androgen receptor’s expression in prostate cancer

Summary: In this study, authors analyzed the link between the presence of CpG methylation in the 5’UTR and promoter regions of AR and loss of AR expression. Their data suggest that AR region 1 genomic availability is crucial for AR expression and the inhibition of histone methyltransferases might hold promise for AR re-expression.

Comments:

1. Introduction: the authors state that “Jarrard et al., showed that treatment with low doses of the demethylating drug 5-aza-2-deoxycytidine (DAC) for a short period of time was able to restore AR mRNA expression in some AR negative cell lines but not in DU-145 cells, in which AR expression was not recovered even after chronic exposure for 2 months”. However according to Figure 2, no AR expression, either at mRNA (data not shown) or protein level (Figure 2D), was observed in DU-145 or PC-3 cells. What the authors think about this discrepant result? The authors should include more AR negative cells to confirm this.

2. Line 94: Nakayama et al., demonstrated that exposure of DU-145 to DAC alone or in combination with Trichostatin A (TSA) led to successful AR transcript expression. However according to figure 3 no significant alteration of AR expression was found upon exposure to DAC+TSA combined treatment in DU-145 cells, both at transcript (data not shown) and protein level (Figure 3C). Again, authors should discuss this discrepant result.

3.What is effect on the methylation profile after DAC and TSA treatment in LNCaP and RWPE-1 cells?

4. Figure 3B: Were these cells also treated with TSA? Please indicate in the figure.

5. Author can include their opinion in discussion on how their findings will help in ADT treatment and overcome castration resistance problems in prostate cancer patients.

6. Figure 2D: PC-3 cells: lane 2 from left, is that a non-specific band?

Minor points:

1.     Line 360 and 387: “Data not shown”; please show these data.

2.     Figure legends: figure numbers are incorrect.

Author Response

Reviewer #1:

Novel insights on the role of epigenetics in androgen receptor’s expression in prostate cancer

Summary: In this study, authors analyzed the link between the presence of CpG methylation in the 5’UTR and promoter regions of AR and loss of AR expression. Their data suggest that AR region 1 genomic availability is crucial for AR expression and the inhibition of histone methyltransferases might hold promise for AR re-expression.

Comments:

1.Introduction: the authors state that “Jarrard et al., showed that treatment with low doses of the demethylating drug 5-aza-2-deoxycytidine (DAC) for a short period of time was able to restore AR mRNA expression in some AR negative cell lines but not in DU-145 cells, in which AR expression was not recovered even after chronic exposure for 2 months”. However according to Figure 2, no AR expression, either at mRNA (data not shown) or protein level (Figure 2D), was observed in DU-145 or PC-3 cells. What the authors think about this discrepant result? The authors should include more AR negative cells to confirm this.

Response: We thank the reviewer’s comments and questions. From literature research, it is possible to verify that studies reporting the baseline expression levels of AR in PC-3 cells are contradictory (for example, studies https://doi.org/10.1016/j.febslet.2006.03.041, https://doi.org/10.1002/pros.2990220103, https://doi.org/10.1016/0039-128X(94)00031-7 and doi: 10.1158/1535-7163.MCT-09-0486), with some authors reporting low or undetectable AR levels, while others were able to detect AR expression in PC-3. Thus, our results regarding absent AR expression in PC-3 cells are supported by previous reports. Moreover, no AR expression was rescued in these cells following DAC treatment. Thus, our data indicate that methylation might not be the mechanism behind AR downregulation in this specific cell line, (doi: 10.1158/1535-7163.MCT-09-0486 and DOI: 10.1093/carcin/bgh310), but instead, other epigenetic mechanisms, particularly, histone methylation associated with lack of expression (DOI: 10.1080/15592294.2016.1146851).

As the reviewer might be aware, contrarily to other models, human PCa cell lines available are limited, and we do not have any additional AR-negative cell line in our Institute.

2.Line 94: Nakayama et al., demonstrated that exposure of DU-145 to DAC alone or in combination with Trichostatin A (TSA) led to successful AR transcript expression. However according to figure 3 no significant alteration of AR expression was found upon exposure to DAC+TSA combined treatment in DU-145 cells, both at transcript (data not shown) and protein level (Figure 3C). Again, authors should discuss this discrepant result.

Response: We thank the reviewer’s comments and questions. Indeed, our findings do not overlap with those obtained by Nakayama and colleagues. However, different results have been reported concerning the effect of DAC treatment alone on DU-145 AR re-expression. For example, Nakayama et al. (PMID: 11140692) showed different results to those obtained by Jarrard et al. (PMID: 9850055). Moreover, the discrepant results following DU-145 DAC+TSA treatment between our study and Nakayama’s may be related with the different experimental conditions, namely in cell culturing passages. Importantly, the enrichment of repressive histone marks at the receptor’s promoter demonstrated by ChiP-PCR is in line with the lack of AR re-expression following combined treatment.

3.What is effect on the methylation profile after DAC and TSA treatment in LNCaP and RWPE-1 cells?

Response: We thank the reviewer’s comments and questions. However, since both cell lines were AR-positive and displayed low methylation levels (Figure 1, Figure S1 and S2), no treatment was performed.

4.Figure 3B: Were these cells also treated with TSA? Please indicate in the figure.

Response: We thank the reviewer’s comments and questions. Indeed, this cell line was treated with both DAC and TSA. However, during the assemble of the figure, the label was misplaced. Figure 3 was corrected (New Figure 3).

5.Author can include their opinion in discussion on how their findings will help in ADT treatment and overcome castration resistance problems in prostate cancer patients.

Response: We thank the reviewer’s comments and questions. Information was added accordingly (line 487-493).

6.Figure 2D: PC-3 cells: lane 2 from left, is that a non-specific band?

Response: We thank the reviewer’s comments and questions. Indeed, the band in question in not the specific AR full length band (110 kDa). As per comparison with the positive control LNCaP AR full length (band’s size was added), the band displays lower molecular weight and, therefore, not specific.

Minor points:

1.Line 360 and 387: “Data not shown”; please show these data.

Response: We thank the reviewer’s comments and questions. The authors decided not to include these data on the manuscript, since no AR PCR amplification was observed, for both treatment conditions. Thus, the plotting of the data would be 0 for both groups and thus not included in the manuscript.

2.Figure legends: figure numbers are incorrect.

Response: We thank the reviewer for pointed the typos. The figures’ numbers were corrected.

Reviewer 2 Report

1.       The work of a Camilo and colleagues is an interesting epigenetic-based study aimed at evaluating the relationship between the presence of CpG methylation in the 5’UTR and promoter regions of androgen receptor and loss of androgen receptor in vitro in a cell model of prostate cancer. Main data indicate the presence of an heterogeneity in the methylation signature of androgen receptor negative cell lines and pinpoint the proximal promoter region as the most consistently methylated site. Thus, the author demonstrated the epigenetic regulation of androgen receptor by promoter methylation. The work is particularly of interest given the importance of the tumor and the limited diagnostic and prognostic options. I therefore recommend considering this manuscript for publication in Biomolecules. Here some comment/minor observation:

2.       The work is interesting and well conducted. The scientific writing is good, and the results are well presented and properly discussed. However, I suggest improving the discussion with details on the study limitations and by including conclusions.

3.       Lines 60-62 or 427-428 as a support, I suggest including this additional reference on the role of improper DNA methylation and cancer development and progression (PMID: 27223861)

4.    androgen receptor is epigenetically regulated by histone prosttranslational modifications. Dysregulations in these pathways have been reported to be related to prostate cancer onset and progression. Histone prosttranslational modifications and DNA methylation are highly connected in regulating the expression of genes. These points should be discussed in relation to the data of the present study, authors can check.

a.       https://www.ncbi.nlm.nih.gov/pmc/articles/PMC10070878/

b.       https://www.ncbi.nlm.nih.gov/pmc/articles/PMC3150559/

c.       https://www.spandidos-publications.com/10.3892/or.2013.2344

d.       https://www.tandfonline.com/doi/10.1128/mcb.00147-06

5.    A stronger effect than DAC might be reached with a more recently developed hypomethylating compound guadecitabine

6.    Section 2.1, 2.3, 2.4, 2.5, 2.6, 2.7 and 2.8 should be improved with supporting references. Methods should be adequately supported by references reporting the employed techniques and protocols

7.    In the methods, the total number of clones for each sample should be detailed

8.    Lines 187-190 this sentence should belong to statistics section (2.9 section)

9.    Line 335, the rationale behind the selection of miR-130a, CCND2 and RASSF1A should be given more in detail. Their promoter were previously reported as improperly methylated in prostate cancer? As a methylation control, I suggest using LINEs  

10.Authors should further consider exposing the cells to DAC for more days, given his effect being frequently studied after 5-7 days of treatment.

English is fine

Author Response

The work of a Camilo and colleagues is an interesting epigenetic-based study aimed at evaluating the relationship between the presence of CpG methylation in the 5’UTR and promoter regions of androgen receptor and loss of androgen receptor in vitro in a cell model of prostate cancer. Main data indicate the presence of a heterogeneity in the methylation signature of androgen receptor negative cell lines and pinpoint the proximal promoter region as the most consistently methylated site. Thus, the author demonstrated the epigenetic regulation of androgen receptor by promoter methylation. The work is particularly of interest given the importance of the tumor and the limited diagnostic and prognostic options. I therefore recommend considering this manuscript for publication in Biomolecules. Here some comment/minor observation:

1.The work is interesting and well conducted. The scientific writing is good, and the results are well presented and properly discussed. However, I suggest improving the discussion with details on the study limitations and by including conclusions.

Response: The requested information was added accordingly (line 494-498 and line 499-501).

2.Lines 60-62 or 427-428 as a support, I suggest including this additional reference on the role of improper DNA methylation and cancer development and progression (PMID: 27223861).

Response: We thank the reviewer’s suggestion. The mentioned reference was added as requested.

3.androgen receptor is epigenetically regulated by histone prosttranslational modifications. Dysregulations in these pathways have been reported to be related to prostate cancer onset and progression. Histone prosttranslational modifications and DNA methylation are highly connected in regulating the expression of genes. These points should be discussed in relation to the data of the present study, authors can check.

  1. https://www.ncbi.nlm.nih.gov/pmc/articles/PMC10070878/
  2. https://www.ncbi.nlm.nih.gov/pmc/articles/PMC3150559/
  3. https://www.spandidos-publications.com/10.3892/or.2013.2344
  4. https://www.tandfonline.com/doi/10.1128/mcb.00147-06

Response: We thank the reviewer’s remarks. Indeed, histone post-translational modifications and DNA methylation frequently work together to regulate gene expression, including in PCa. This point was addressed accordingly (lines 479-483)

  1. A stronger effect than DAC might be reached with a more recently developed hypomethylating compound guadecitabine

Response: We thank the reviewer’s suggestion. As depicted in Figure 2C, DAC successfully decreased the methylation levels of known PCa downregulated genes (miR-130a, CCDN2 and RASSF1A). Thus, DAC’s demethylating capacity was not compromised. However, from our data, we can suggest that downregulation in DU-145 seems to be a highly controlled process that encompasses other epigenetic mechanisms, and AR re-expression is not solely dependent on methylation.

  1. Section 2.1, 2.3, 2.4, 2.5, 2.6, 2.7 and 2.8 should be improved with supporting references. Methods should be adequately supported by references reporting the employed techniques and protocols

Response: We thank the reviewer’s comments. The manuscript was altered, accordingly.

  1. In the methods, the total number of clones for each sample should be detailed

Response: We thank the reviewer’s comments and questions. The authors have altered accordingly.

  1. Lines 187-190 this sentence should belong to statistics section (2.9 section)

Response: The alteration was performed (lines 262-265).

  1. Line 355, the rationale behind the selection of miR-130a, CCND2 and RASSF1A should be given more in detail. Their promoter were previously reported as improperly methylated in prostate cancer? As a methylation control, I suggest using LINEs

Response: We thank the reviewer’s comments and questions. Herein, since no methylation alterations were observed in AR promoter, we tested miR-130a, CCDN2 and RASSF1A methylation levels as proof-of-principle to evaluate DAC treatment efficacy, as we have previously demonstrated in prostate cancer (Cancer Lett. 10.1016/j.canlet.2016.10.028; J Mol Med10.1007/s00109-006-0099-4 and Clin Epigenetics. 10.1186/s13148-019-0779-x). Nevertheless, the authors acknowledge the suggestion of using LINEs as methylation control.

10.Authors should further consider exposing the cells to DAC for more days, given his effect being frequently studied after 5-7 days of treatment.

Response: We thank the reviewer’s comments and questions. Although we have tested different treatment conditions and time points, no differences were apparent among the tested experimental settings. Therefore, the less time and drug-consuming experimental condition (3 days) was selected for further studies.

Reviewer 3 Report

The manuscript titled "Novel insights on the role…. Prostate cancer," by Camilo et al., demonstrates the heterogeneity in the methylation signature of AR negative cell lines. The data is broadly convincing, with few concerns that must be addressed before the manuscript can be published.

Major Concerns:

1.       The study's strength is using multiple cell lines to demonstrate the methylation pattern in the AR promoter region.

2.       There are two Figures: 1. This error needs to be rectified in the figure and text.

3.       How was the concentration of DAC (5µM) and TSA (0.33µM) determined?  

4.       How was the overall percentage of methylation determined? How the black and white circles were derived in Figures 1 and 2) (Two figures are labeled 1).

5.       It would be prudent if authors make a table showing percentage methylation marks in AR promoters in different cell lines.

Author Response

The manuscript titled "Novel insights on the role…. Prostate cancer," by Camilo et al., demonstrates the heterogeneity in the methylation signature of AR negative cell lines. The data is broadly convincing, with few concerns that must be addressed before the manuscript can be published.

Major Concerns:

1.The study's strength is using multiple cell lines to demonstrate the methylation pattern in the AR promoter region.

Response: We thank the reviewer’s positive comments.

2.There are two Figures: 1. This error needs to be rectified in the figure and text.

Response: We thank the reviewer’s remark. The error was corrected.

3.How was the concentration of DAC (5µM) and TSA (0.33µM) determined? 

Response: We thank the reviewer’s comments and questions. DAC and TSA concentrations were selected based on literature reports (references 26 and 27) and in house pre-screening experiments.

  1. How was the overall percentage of methylation determined? How the black and white circles were derived in Figures 1 and 2) (Two figures are labeled 1).

Response: We thank the reviewer’s comments and questions. The methylation status of each CG, per clone, was determined by analysing the sequencing results on QUMA software. Thus, black circles represent methylated CGs, whereas white ones represent unmethylated CGs (Figure S2-4). The schemes on Figures 1 and 2 represent the mean % of the 15 analysed clones that were methylated for each CG. The circles were then assembled in excel (% of methylation in black; % of non-methylated CGs in white).

  1. It would be prudent if authors make a table showing percentage methylation marks in AR promoters in different cell lines

Response: We thank the reviewer’s comments and questions. The percentage of methylation, per CG, in each AR promoter regions, was calculated and provided in Figure 1. Only the values for the most relevant cell lines for the study are shown. The other tested cell lines were included to show the broad AR methylation spectrum, per CG, and not for quantification of respective methylation after treatment.

Round 2

Reviewer 3 Report

Thank you for addressing the concerns. 

Author Response

Dear Editor and reviewer,
thanks for the comments.
The required information was added to the manuscript.
Thank you so much for your attention.
Sincerely,
Carmen Jerónimo
